# Simply crushed zizyphi spinosi semen prevents neurodegenerative diseases and reverses age-related cognitive decline in mice

**Tomohiro Umeda[1,2], Ayumi Sakai[1,2], Rumi Uekado[1], Keiko Shigemori[1], Ryota Nakajima[3], Kei Yamana[3], Takami Tomiyama[1,2]***

[1]Department of Translational Neuroscience, Osaka Metropolitan University Graduate School of Medicine, Osaka, Japan; [2]Cerebro Pharma Inc, Osaka, Japan; [3]NOMON Co., Ltd, and New Business Development Unit, Teijin Ltd, Kasumigaseki Common Gate West Tower, Tokyo, Japan

## eLife assessment

The authors made a **useful** finding that Zizyphi spinosi semen, a traditional Chinese medicine, has demonstrated excellent biological activity and potential therapeutic effects against Alzheimer's disease (AD). The researchers presented the effects, but the research evidence for the mechanism was **incomplete**. The main claims were only partially supported.

*For correspondence:
tomi@omu.ac.jp

**Abstract** Neurodegenerative diseases are age-related disorders characterized by the cerebral accumulation of amyloidogenic proteins, and cellular senescence underlies their pathogenesis. Thus, it is necessary for preventing these diseases to remove toxic proteins, repair damaged neurons, and suppress cellular senescence. As a source for such prophylactic agents, we selected zizyphi spinosi semen (ZSS), a medicinal herb used in traditional Chinese medicine. Oral administration of ZSS hot water extract ameliorated Aβ and tau pathology and cognitive impairment in mouse models of Alzheimer's disease and frontotemporal dementia. Non-extracted ZSS simple crush powder showed stronger effects than the extract and improved α-synuclein pathology and cognitive/motor function in Parkinson's disease model mice. Furthermore, when administered to normal aged mice, the ZSS powder suppressed cellular senescence, reduced DNA oxidation, promoted brain-derived neurotrophic factor expression and neurogenesis, and enhanced cognition to levels similar to those in young mice. The quantity of known active ingredients of ZSS, jujuboside A, jujuboside B, and spinosin was not proportional to the nootropic activity of ZSS. These results suggest that ZSS simple crush powder is a promising dietary material for the prevention of neurodegenerative diseases and brain aging.

## Introduction

Cerebral accumulation of amyloidogenic proteins is a hallmark of neurodegenerative diseases; Aβ and tau accumulate in Alzheimer's disease (AD), tau accumulates in frontotemporal dementia (FTD), and α-synuclein accumulates in Parkinson's disease (PD) and dementia with Lewy bodies (DLB). In patients' brains, these proteins aggregate into toxic oligomers and fibrils to induce synaptic dysfunction (*Li and Selkoe, 2020*; *Gutierrez and Limon, 2022*) and intercellular propagation of neuropathology (*Jaunmuktane and Brandner, 2020*; *Umeda et al., 2022*). In consequence, cognitive function declines in

AD, FTD, and DLB, and motor function is impaired in PD. The accumulation of these proteins begins decades before the disease onset and many neurons die by the time clinical symptoms emerge (*Jack et al., 2013*; *Ossenkoppele et al., 2022*). These findings indicate the importance of early diagnosis and prevention and that prophylactic agents for neurodegenerative diseases should have activities to remove toxic oligomers and repair damaged neurons.

Neurodegenerative diseases are age-related disorders (*Wyss-Coray, 2016*; *Hou et al., 2019*), and recent evidence suggests that cellular senescence underlies their pathogenesis (*Sahu et al., 2022*; *Shafqat et al., 2023*). Cellular senescence is a physiological phenomenon observed in aging in which proliferating cells undergo stable cell cycle arrest. This phenomenon is induced by telomere attrition, epigenetic alterations, oxidative stress, mitochondrial dysfunction, mechanical or shear stress, pathogens, and activation of oncogenes (*Shafqat et al., 2023*). In neurodegenerative diseases, abnormal protein accumulation triggers reactive oxygen species (ROS) generation which causes oxidative stress and organelle dysfunction leading to cellular senescence (*Sahu et al., 2022*). Senescent cells accelerate the aging and damage of surrounding tissues through senescence-associated secretory phenotype composed of cytokines, chemokines, and growth factors (*Sahu et al., 2022*; *Shafqat et al., 2023*; *González-Gualda et al., 2021*), which in turn exacerbates neurodegenerating process. Eliminating senescent cells with senolytic drugs, such as dasatinib+quercetin (D+Q), has been shown to ameliorate neuropathology, inflammation, and cognitive function in AD model mice (*Zhang et al., 2019*). Thus, cellular senescence is another target to prevent neurodegenerative diseases.

The current mainstream in drug development for neurodegenerative diseases is immunotherapy. Two Aβ antibodies have been approved, and tau and α-synuclein antibodies along with other Aβ antibodies are also in clinical trials. However, these drugs have problems in terms of cost, safety, and invasiveness. They are generally very expensive, and their side effects are often serious. In addition, patients must receive intravenous treatment at a hospital, which is a burden to the patient. Such treatments are not suitable for long-term prevention. Prophylactic agents must be safe, inexpensive, and noninvasively available so that all asymptomatic and undiagnosed people can take them for a long period at their own discretion. Also, they should be broadly effective against etiologic proteins, capable of repairing neurons, and effective at suppressing cellular senescence. These requirements are difficult to meet with single-ingredient pharmaceuticals, but it may be feasible by taking proper diets composed of multiple ingredients.

In search of materials for such diets, we explored medicinal herbs used in traditional Chinese medicine, where several herbs such as *Rehmanniae radix* (*Jia et al., 2023*), *Polygalae radix* (*Zhao et al., 2020*), ziziphi spinosi semen (ZSS) (*Zhang et al., 2022*), and *Acorus tatarinowii/gramineus rhizome* (*Kim et al., 2022*) are claimed to be effective for amnesia. In the present study, we focused on ZSS, the seeds of *Ziziphus jujuba Miller* var. *spinosa*, because ZSS is treated as a non-pharmaceutical in Japan but the other herbs are regarded as pharmaceuticals, i.e., not suitable for diets. In traditional Chinese medicine, medicinal herbs are often subjected to decoction; the resultant extracts are taken as medicines but the extraction residues are usually discarded. However, our previous studies revealed that the residues also have some functional ingredients and that hot water extraction tends to lose some volatile substances by evaporation and possibly break down some heat-sensitive components, which led us to conclude that simple crushing without extraction is a better way to maximize the effects of medicinal herbs (*Umeda et al., 2024b*; *Umeda et al., 2024a*). Based on these observations, we made three preparations from dried ZSS: hot water extract, extraction residue, and non-extracted simple crush powder, and examined their efficacy in three different mouse models of neurodegenerative diseases. In addition, we evaluated the antiaging effects of ZSS in normal aged mice. Our results show that non-extracted ZSS simple crush powder meets all the requirements for neurodegenerative disease-prophylactic agents and that the ZSS powder is a promising dietary material against neurodegenerative diseases and brain aging.

## Results
### Effects of ZSS hot water extract on APP23 mice

Initially, we examined the effects of ZSS hot water extract on cognitive function and Aβ pathology in AD model mice. APP23 mice express human APP with the Swedish (KM670/671NL) mutation (*Sturchler-Pierrat et al., 1997*) and show Aβ oligomer accumulation, synapse loss, memory impairment, and

amyloid deposition by 15 months (*Umeda et al., 2021b*). ZSS extract was orally administered to 13- to 15-month-old APP23 mice (mean body weight, 28.9 g) at 0.1 mg/shot for 1 month. Cognitive function of mice was assessed by the Morris water maze test. The treatment improved mouse memory, but the effect was not complete (*Figure 1A*). We repeated the experiment with a higher dose of ZSS extract in 15- to 16-month-old mice (mean body weight, 28.9 g). 0.5 mg/shot significantly improved mouse memory to a level similar to that of non-transgenic (non-Tg) littermates (*Figure 1B*). Aβ pathology was assessed by immunohistochemistry in mice that received the lower dose (0.1 mg/shot). ZSS extract significantly reduced the levels of Aβ oligomers (*Figure 1C*) and amyloid deposits (*Figure 1D*) in the cerebral cortex and hippocampus. Synaptophysin levels in the mossy fibers of hippocampal CA2/3 regions were significantly recovered by the treatment (*Figure 1E*).

## Effects of ZSS hot water extract on Tau784 mice

Next, we tested the effects of ZSS hot water extract on cognitive function and tau pathology in FTD model mice. Tau784 mice express both 3-repeat and 4-repeat human tau with the dominant expression of 4-repeat human tau at adult age by the presence of a tau intron mutation (*Umeda et al., 2013*). They exhibit tau hyperphosphorylation, tau oligomer accumulation, synapse loss, and memory impairment at 6 months (*Umeda et al., 2013*; *Umeda et al., 2015*). ZSS extract was orally administered to 14-month-old Tau784 mice (mean body weight, 35.1 g) at 0.1 and 0.5 mg/shot for 1 month. Mouse memory was improved in a dose-dependent manner; the higher dose achieved a complete recovery, reaching a level similar to that of non-Tg littermates, whereas the lower dose showed only a moderate effect (*Figure 2A*). Tau pathology was examined by immunohistochemistry. The levels of phosphorylated tau in the hippocampus and tau oligomers in the cerebral cortex were significantly reduced by the treatment in a dose-dependent manner (*Figure 2B*). Synaptophysin levels in the hippocampal CA2/3 regions were also significantly recovered in a dose-dependent manner (*Figure 2C*).

## Comparison of ZSS hot water extract, extraction residue, and non-extracted simple crush powder in Tau784 mice

Next, we compared the effects of the hot water extract, extraction residue, and non-extracted simple crush powder of ZSS in Tau784 mice. The hot water extract and simple crush powder were administered to 8- to 12-month-old mice (mean body weight, 29.3 g) at 0.1 mg/shot for 1 month. The hot water extract improved mouse memory, but the effect was incomplete (*Figure 3A*). In contrast, the simple crush powder markedly enhanced mouse memory to a level even higher than that of non-Tg littermates. Subsequently, the simple crush powder and extraction residue were administered to 8- to 11-month-old mice (mean body weight, 31.2 g) at 0.1 mg/shot for 1 month. Again, the simple crush powder displayed a strong effect on mouse memory (*Figure 3B*). On the other hand, the extraction residue showed only a moderate effect, similar to that of the hot water extract. The levels of phosphorylated tau in the hippocampus (*Figure 3C*) and tau oligomers in the cerebral cortex (*Figure 3D*) were significantly attenuated by all three preparations, with the simple crush powder showing the strongest effects. Synaptophysin levels in the hippocampal CA2/3 regions were significantly recovered by the simple crush powder to a level higher than that in non-Tg littermates, whereas the hot water extract and extraction residue showed only slight effects (*Figure 3E*). To evaluate the ability of ZSS to repair damaged neurons, we examined the expression level of brain-derived neurotrophic factor (BDNF), which promotes the growth and regeneration of neurons (*Colucci-D'Amato et al., 2020*). The simple crush powder significantly increased BDNF expression in the hippocampus to a level even higher than that in non-Tg littermates (*Figure 3F*). In contrast, the hot water extract and extraction residue showed only slight effects on BDNF.

## Effects of ZSS simple crush powder on Huα-Syn(A53T) mice

Next, we examined the effects of ZSS simple crush powder on motor function and α-synuclein pathology in PD model mice. Huα-Syn(A53T) mice express human α-synuclein with A53T mutation (*Lee et al., 2002*) and exhibit α-synuclein phosphorylation, α-synuclein oligomer accumulation, synapse loss, and memory impairment at 6 months, and motor dysfunction at 9 months, indicating that they can be used as a model of DLB at 6–8 months and as a model of PD after 9 months (*Umeda et al., 2021a*). ZSS powder was orally administered to 8-month-old mice (mean body weight, 29.8 g) at 0.1 mg/shot for 1 month. Motor function of mice was assessed by the rotarod test. The treatment significantly improved

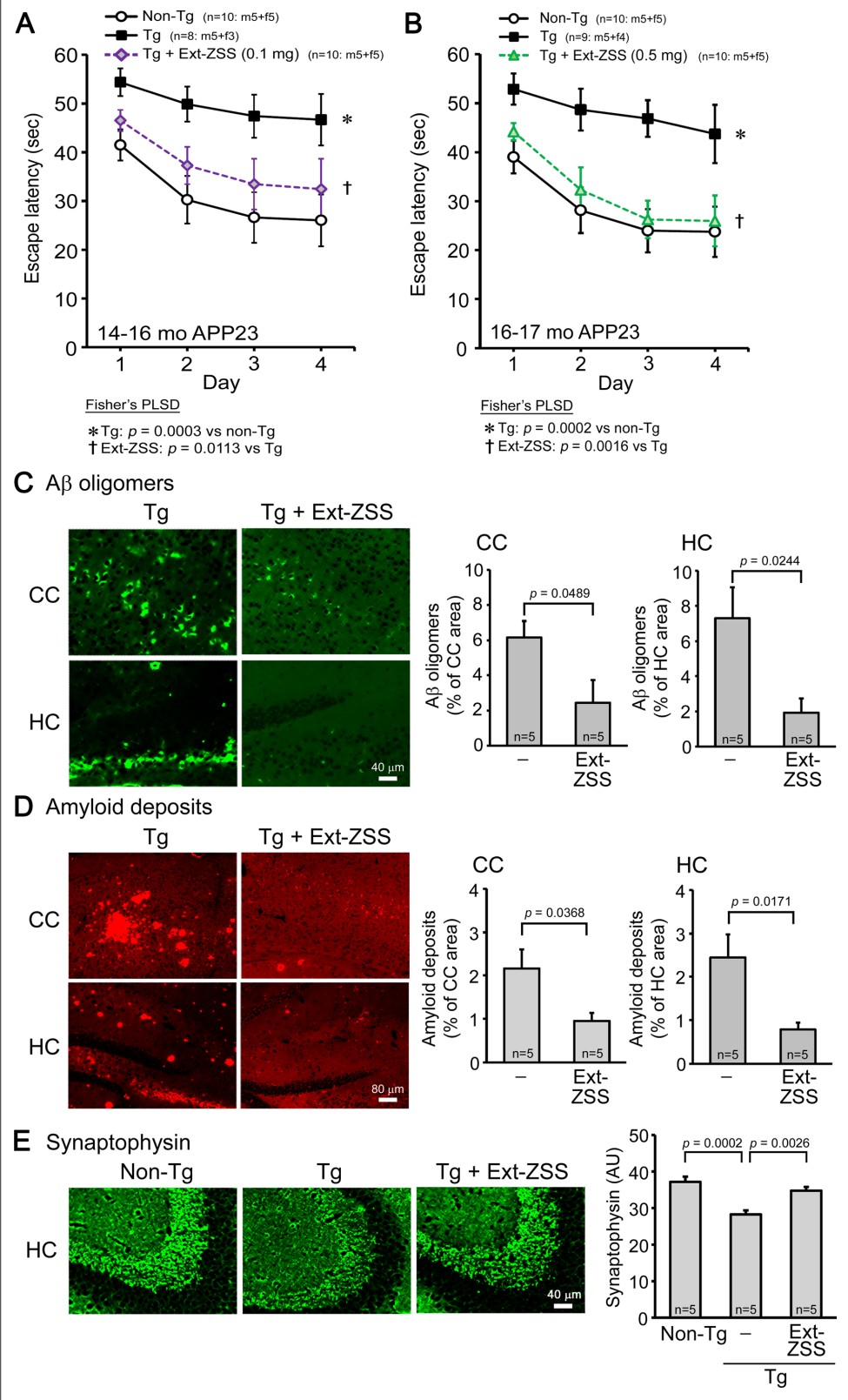

**Figure 1.** Effects of zizyphi spinosi semen (ZSS) hot water extract on APP23 mice. Hot water extract of ZSS (Ext-ZSS) was administered to 13- to 15-month-old and 15- to 16-month-old APP23 mice at 0.1 and 0.5 mg/shot, respectively, for 1 month. (**A**) Ext-ZSS at 0.1 mg/shot improved mouse memory, but its effect was not complete. (**B**) Ext-ZSS at 0.5 mg/shot significantly improved mouse memory to a level similar to that of non-Tg littermates.

*Figure 1 continued on next page*

*Figure 1 continued*

(**C**) Aβ pathologies were assessed in mice that received the lower dose. Ext-ZSS at 0.1 mg/shot significantly reduced the levels of Aβ oligomers in the cerebral cortex (CC) and hippocampus (HC). (**D**) Amyloid deposits in these regions were also significantly decreased with Ext-ZSS. (**E**) Ext-ZSS significantly restored synaptophysin levels in the mossy fibers of hippocampal CA2/3 regions to a level similar to that of non-Tg littermates. AU, arbitrary unit. Each point and bar represent the mean ± SEM. The numbers of total, male, and female mice analyzed are shown in each figure as n=x: m (male)+f (female).

motor function of Tg mice to a level similar to that of non-Tg littermates (*Figure 4A*). α-Synuclein pathology was assessed by immunohistochemistry. The levels of phosphorylated α-synuclein and α-synuclein oligomers in the hippocampus were significantly reduced by the treatment (*Figure 4B*). BDNF expression was examined in the cerebral cortex and substantia nigra; the latter brain region is particularly vulnerable to α-synuclein-induced neurodegeneration, leading to the motor dysfunction in PD. ZSS powder significantly enhanced BDNF expression in both brain regions (*Figure 4C*). Neurogenesis plays an important role in learning and memory and is shown to decrease in chronic stress, aging, and neurodegenerative diseases (*Horgusluoglu et al., 2017*). Thus, we examined neurogenesis in the dentate gyrus and substantia nigra. Neurogenesis in Tg mice showed a tendency to decrease in both brain regions (*Figure 4D*). ZSS powder significantly increased neurogenesis to levels higher than those in non-Tg littermates. These results suggest that ZSS simple crush powder is effective at repairing damaged neurons. As mentioned above, Huα-Syn(A53T) mice can be used as a model of DLB at 6–8 months. Thus, we investigated the effect of ZSS powder on DLB using younger mice. ZSS powder was orally administered to 6- to 7-month-old mice (mean body weight, 28.0 g) at 0.1 mg/shot for 1 month. Mouse memory was significantly improved to a level similar to or slightly less than that of non-Tg littermates (*Figure 4E*).

Taken together, these results suggest that ZSS simple crush powder is a promising dietary material for preventing neurodegenerative diseases.

## Effects of ZSS simple crush powder on normal aged mice

The finding that Tau784 mice treated with ZSS simple crush powder displayed higher cognitive function than age-matched non-Tg littermates (*Figure 3A and B*) led us to speculate that the powder has not only disease-preventing effects but also brain-rejuvenating effects. Thus, we studied the antiaging effects of ZSS powder using two wild-type groups, aged and young mice. ZSS powder was orally administered to aged, 16- to 18-month-old (mean body weight, 36.1 g), and the younger, 8-month-old C57BL/6 mice (mean body weight, 31.1 g) at 0.1 mg/shot for 1 month. Their cognitive function and levels of synaptophysin, BDNF expression, and neurogenesis were compared with those in water-administered aged and young wild-type mice. As shown in *Figure 5A*, cognitive function of aged mice was significantly lower than that of young mice. ZSS powder significantly enhanced aged mice's cognition to a level similar to that of water-treated young mice. An enhanced memory was also observed in ZSS-treated young mice, but it was not significant. Synaptophysin levels in the hippocampal CA2/3 regions (*Figure 5B*), BDNF expression in the hippocampus (*Figure 5C*), and neurogenesis in the dentate gyrus (*Figure 5D*) were significantly increased in ZSS-treated aged mice, reaching levels similar to those in water-treated young mice. Increased levels of synaptophysin and BDNF were also observed in ZSS-treated young mice, but the effects were not significant. Cellular senescence is one of the hallmarks of aging (*López-Otín et al., 2023*) and has been suggested to underlie the pathogenesis of neurodegenerative diseases (*Sahu et al., 2022*; *Shafqat et al., 2023*). We measured the levels of cellular senescence focusing on its intracellular markers, p16[INK4a], p21[CIP1/WAF1], and γH2AX; the former two represent cell cycle arrest and the last one reflects DNA damage (*Sahu et al., 2022*; *González-Gualda et al., 2021*). The levels of these markers in the cerebral cortex of aged mice were significantly higher than those in young mice (*Figure 5E*). ZSS powder significantly reduced them to levels similar to or lower than those in water-treated young mice (*Figure 5E*). Reduced cellular senescence was also observed for p21 and γH2AX in ZSS-treated young mice, but the effects were not significant.

Cellular senescence is induced by DNA damage triggered by various stressors, and oxidative stress is a major cause of DNA damage (*Nousis et al., 2023*). To evaluate the antioxidant effect of ZSS powder, we measured the levels of 8-hydroxy-2'-deoxyguanosine (8-OHdG), a marker of DNA

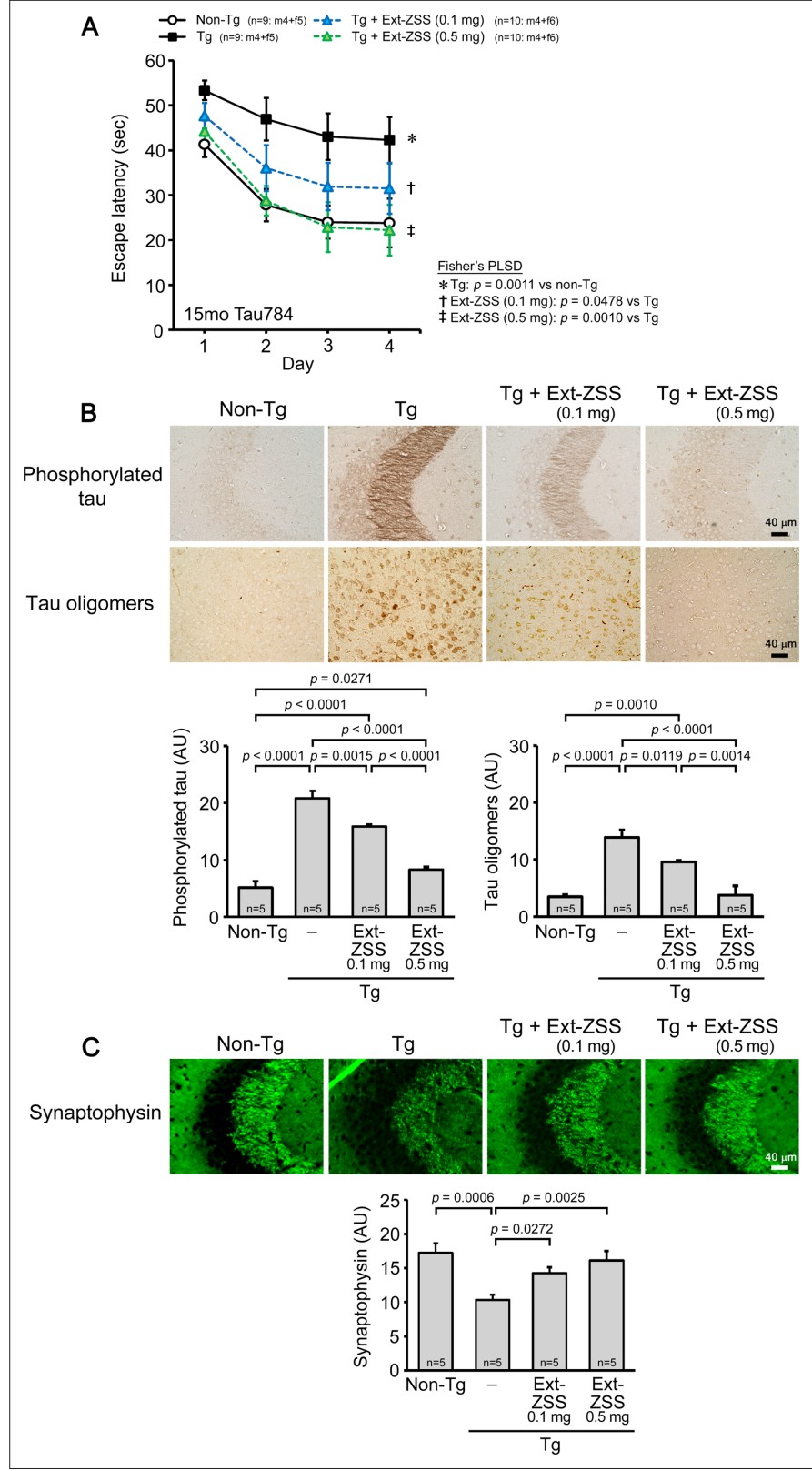

**Figure 2.** Effects of zizyphi spinosi semen (ZSS) hot water extract on Tau784 mice. Hot water extract of ZSS (Ext-ZSS) was administered to 14-month-old Tau784 mice at 0.1 and 0.5 mg/shot for 1 month. (**A**) Ext-ZSS improved mouse memory in a dose-dependent manner; the higher dose achieved a complete recovery to a level similar to that of non-Tg littermates. (**B**) Ext-ZSS significantly reduced the levels of phosphorylated tau in the hippocampus

*Figure 2 continued on next page*

*Figure 2 continued*

and tau oligomers in the cerebral cortex in a dose-dependent manner. (**C**) The levels of synaptophysin in the hippocampal CA2/3 regions were significantly recovered in a dose-dependent manner. Each point and bar represent the mean ± SEM. The numbers of total, male, and female mice analyzed are shown in each figure as n=x: m (male)+f (female).

oxidation (*Valavanidis et al., 2009*), and its autoantibodies (*Ahmad et al., 2024*) in the brains of ZSS-treated mice. The levels of 8-OHdG and its autoantibodies were both increased in aged mice, where the increase in autoantibodies was more pronounced than that in 8-OHdG (*Figure 6A*). This may be because when DNA oxidation products are continuously produced with aging, autoimmune response is strongly induced to remove them, and as a result, the increase in 8-OHdG is suppressed. Such a persistent immune reaction may cause chronic inflammation leading to cellular senescence. ZSS powder significantly reduced the levels of 8-OHdG and its autoantibodies to levels similar to or lower than those in water-treated young mice. We further examined the radical-scavenging ability of ZSS powder in a cell-free system. ZSS powder quenched superoxide only weakly (*Figure 6B*); its activity was about 1/100 of that of Mamaki leaf powder, which is known to have strong antioxidant activity (*Kartika et al., 2007*). These results suggest that ZSS powder suppresses cellular senescence, at least in part, through its antioxidant action, but this effect is unlikely due to the direct action of ZSS components on free radicals.

## Analysis of the three components of ZSS preparations and their effects on Tau784 mice

The major ingredients of ZSS are jujuboside A, jujuboside B, and spinosin (*Liu et al., 2007*; *Zhang et al., 2014*; *Hua et al., 2022*), all of which have been reported to possess neuroprotective effects (*Zhang et al., 2018*; *Tabassum et al., 2019*; *Liu et al., 2014*; *Jin et al., 2023*; *Ko et al., 2015*; *Lee et al., 2016*; *Xu et al., 2019*). We speculated that ZSS simple crush powder has stronger effects than the hot water extract because the powder contains higher amounts of these components than the extract. To test this possibility, we analyzed the amounts of these components in the hot water extract and simple crush powder of ZSS. Contrary to our expectation, the powder contained less of these substances than the extract (*Table 1*). To more precisely evaluate the contribution of jujuboside A, jujuboside B, and spinosin on mouse cognition, we combined these compounds in water to final contents corresponding to those in 0.5 mg ZSS extract. The mixture, which contained 0.455 μg jujuboside A, 0.2 μg jujuboside B, and 0.7 μg spinosin in 300 μL, was administered to 13- to 16-month-old Tau784 mice (mean body weight, 31.9 g) for 1 month. This treatment showed a much weaker effect on mouse memory (*Figure 7*) than that of 0.5 mg ZSS extract (*Figure 2A*). These results suggest that ZSS contains other active substances besides jujuboside A, jujuboside B, and spinosin, and a significant portion of them may be lost during hot water extraction.

## Discussion

In the present study, we demonstrated that ZSS, particularly its non-extracted simple crush powder, has remarkable medicinal effects on neurodegenerative diseases. It removed Aβ, tau, and α-synuclein oligomers, restored synaptophysin levels, enhanced BDNF expression and neurogenesis, and improved cognitive and motor function in mouse models of AD, FTD, DLB, and PD. Furthermore, in normal aged mice, ZSS powder reduced DNA oxidation and cellular senescence, increased synaptophysin, BDNF, and neurogenesis, and enhanced cognition to levels similar to those in young mice. We proposed that neurodegenerative disease-prophylactic agents should have activities to remove toxic oligomers of etiologic proteins, repair damaged neurons, and suppress cellular senescence. Our results show that the ZSS powder meets these requirements. In addition, such prophylactic agents must be safe, inexpensive, and noninvasively available because the preventive treatment would last for a long period. Since ZSS is safe (i.e. treated as a non-pharmaceutical in Japan), cheaper than medicines, and orally available, the powder can meet these demands as well. Thus, ZSS simple crush powder is a promising dietary material for aged people to avoid neurodegenerative diseases and brain aging (*Figure 8*).

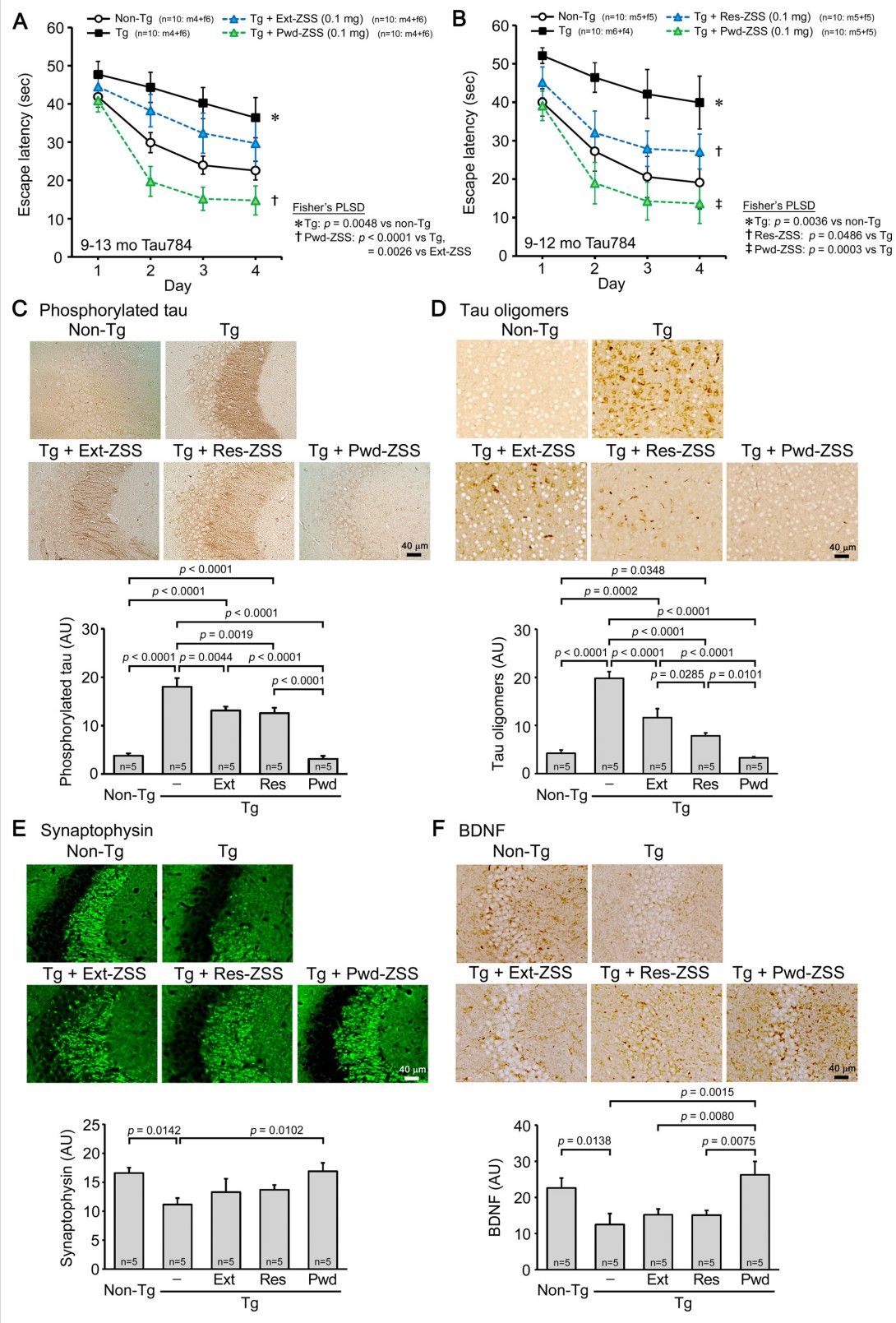

**Figure 3.** Comparison of zizyphi spinosi semen (ZSS) hot water extract, extraction residue, and non-extracted simple crush powder in Tau784 mice. Hot water extract (Ext-ZSS), non-extracted simple crush powder (Pwd-ZSS), and extraction residue (Res-ZSS) of ZSS were administered to 8- to 12-month-old and 8- to 11-month-old Tau784 mice at 0.1 mg/shot for 1 month. (**A**) Pwd-ZSS markedly enhanced mouse memory to a level even higher than that of non-Tg littermates, whereas Ext-ZSS showed only a moderate effect. (**B**) Res-ZSS also showed a moderate effect, similar to that of Ext-ZSS. (**C, D**) The

*Figure 3 continued on next page*

Figure 3 continued

levels of phosphorylated tau in the hippocampus and tau oligomers in the cerebral cortex were significantly attenuated by all three preparations, with Pwd-ZSS showing the strongest effects. (**E**) Pwd-ZSS significantly increased the levels of synaptophysin in the hippocampal CA2/3 region to a level similar to or higher than those in non-Tg littermates. Ext-ZSS and Res-ZSS showed only slight effects. (**F**) Pwd-ZSS significantly enhanced brain-derived neurotrophic factor (BDNF) expression in the hippocampus to a level even higher than that in non-Tg littermates. Ext-ZSS and Res-ZSS showed only slight effects. Each point and bar represent the mean ± SEM. The numbers of total, male, and female mice analyzed are shown in each figure as n=x: m (male)+f (female).

Traditional Chinese medicines are generally formulated as a combination of several herbs. This is because combining multiple herbs can increase efficacy, reduce side effects, and create new synergy actions. In addition, traditional Chinese medicine aims to cure disease by balancing the body rather than targeting specific symptoms or molecules, which could be efficiently achieved by combining various herbal medicines. For example, in Japan, a Chinese medicine called Yokukansan (*Ikarashi and Mizoguchi, 2016*) is often prescribed to relieve the peripheral symptoms of dementia, and it is composed of seven kinds of herbs not including ZSS. As for ZSS, it is prescribed in combination with other herbs to stabilize the mind and cure palpitation, anxiety, and insomnia, and improve amnesia (*Zhang et al., 2022*; *Hua et al., 2022*). However, in order to facilitate the procurement of raw materials and reduce costs, it is better to use fewer herbs as long as the same effect can be obtained. Our results show that ZSS ameliorates core symptoms of dementia by clearing toxic protein oligomers and repairing damaged neurons, without being combined with other herbs. The peripheral symptoms of dementia, known as BPSD (behavioral and psychological symptoms of dementia), include hallucination, delusion, depression, wandering, and agitation. Since ZSS is also effective on insomnia, depression, and anxiety (*Zhang et al., 2022*; *Hua et al., 2022*), ZSS could potentially be used alone for both the core and peripheral symptoms of dementia.

Among the three preparations of ZSS we tested, the simple crush powder displayed the strongest effects, and the extraction residue showed the same efficacy as the hot water extract. We had observed a similar tendency with other herbs, Mamaki and *Acorus tatarinowii/gramineus* leaves (*Umeda et al., 2024b*; *Umeda et al., 2024a*). These results support our previous hypothesis that hot water extraction cannot necessarily recover all active ingredients of medicinal herbs, rather, some functional components will be lost during the process. Identifying true active ingredients would be useful not only for the development of functional foods but also for that of pharmaceuticals to prevent neurodegenerative diseases. ZSS contains various ingredients, and several major components have been identified: jujuboside A, jujuboside B, and spinosin (*Liu et al., 2007*; *Zhang et al., 2014*; *Hua et al., 2022*). These substances have been shown to restore cognitive function in disease model mice. For example, oral administration of jujuboside A promotes Aβ clearance and improves cognition in APP/PS1 mice (*Zhang et al., 2018*). Furthermore, its intracerebroventricular injection prevents sleep loss-induced memory impairment in APP/PS1 mice (*Tabassum et al., 2019*) and ameliorates cognitive impairment in Aβ42-injected mice by reducing the level of Aβ42 (*Liu et al., 2014*). Jujuboside B, when administered intraperitoneally, suppresses febrile seizure in lipopolysaccharide-injected mice (*Jin et al., 2023*). Oral administration of spinosin improves the cognition of Aβ42 oligomer-injected mice (*Ko et al., 2015*) and in adult mice increasing neurogenesis as well as the levels of BDNF (*Lee et al., 2016*). Furthermore, its intracerebroventricular injection attenuates cognitive impairment and restores the levels of BDNF in Aβ42-injected mice (*Xu et al., 2019*). Based on these literatures, we compared the content of jujuboside A, jujuboside B, and spinosin in our ZSS preparations. Despite that the effects of simple crush powder were stronger than that of hot water extract (*Figure 3*), the content of the three compounds in the powder was lower than that in the extract (*Table 1*). In addition, while ZSS extract sufficiently improved mouse memory at 0.5 mg/shot (*Figure 2A*), a mixture of jujuboside A, jujuboside B, and spinosin failed to do so at doses equivalent to their amounts in 0.5 mg ZSS extract (*Figure 7*). These results suggest that the active components of ZSS are other than jujuboside A, jujuboside B, or spinosin. In aged mice, ZSS powder reduced the levels of 8-OHdG and its autoantibodies in the brain (*Figure 6A*); nevertheless, the radical-scavenging ability of ZSS powder was very weak (*Figure 6B*), suggesting that the antioxidant activity of ZSS powder is not due to the direct action of ZSS components on free radicals. It has been shown that plant-derived dietary fibers can make the intestinal environment favorable for gut microbes and consequently affect brain function (*Puhlmann and de Vos, 2022*; *Cuervo-Zanatta et al., 2023*). The effects of ZSS

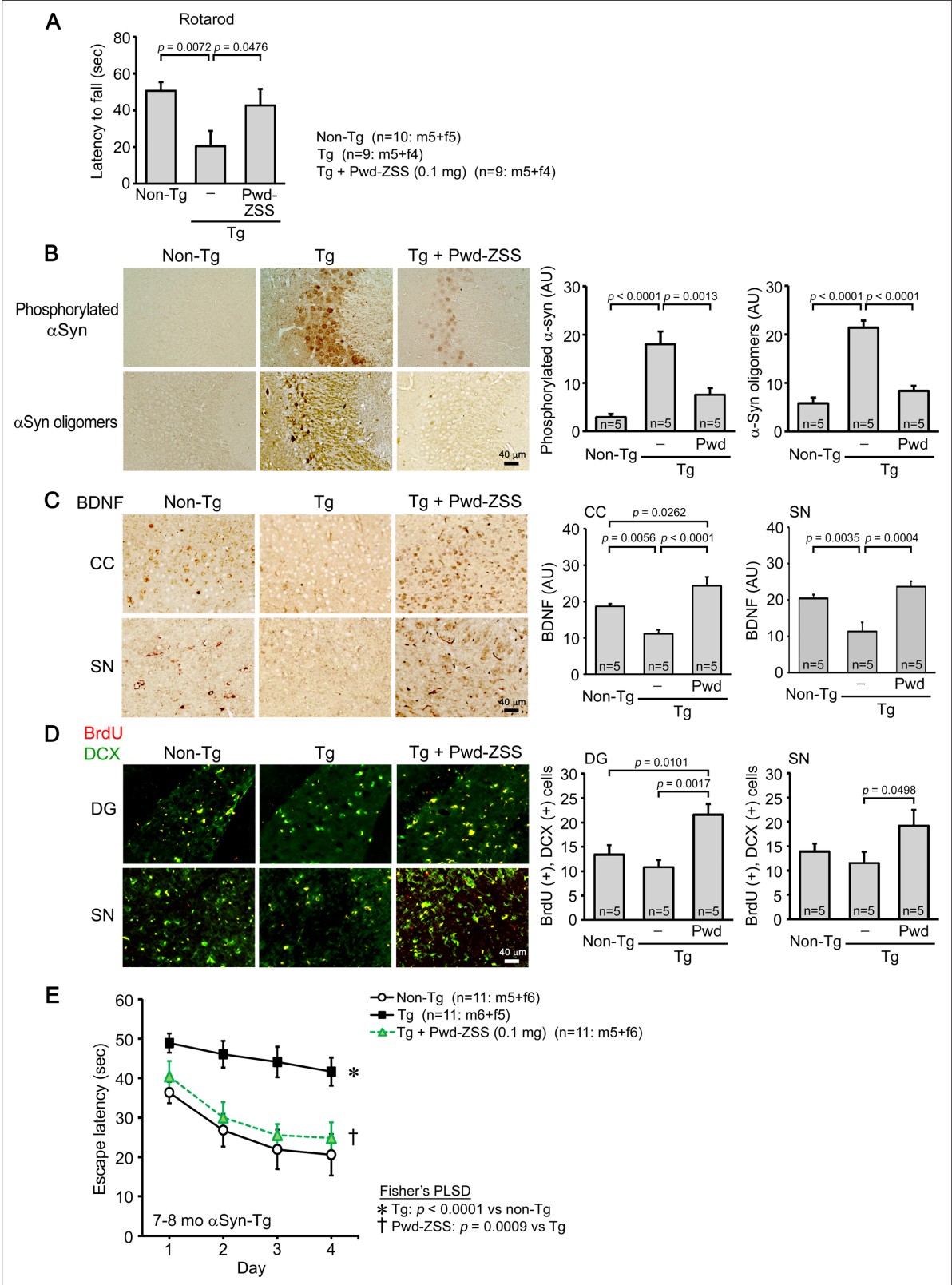

**Figure 4.** Effects of zizyphi spinosi semen (ZSS) simple crush powder on Huα-Syn(A53T) mice. Non-extracted simple crush powder of ZSS (Pwd-ZSS) was administered to 8-month-old Huα-Syn(A53T) mice at 0.1 mg/shot for 1 month. (**A**) Pwd-ZSS significantly improved motor function to a level similar to that of non-Tg littermates. (**B**) Pwd-ZSS significantly reduced the levels of phosphorylated α-synuclein and α-synuclein oligomers in the hippocampus. (**C**) Pwd-ZSS significantly enhanced brain-derived neurotrophic factor (BDNF) expression in the cerebral cortex (CC) and substantia nigra (SN) to a

*Figure 4 continued on next page*

*Figure 4 continued*

level even higher than that in non-Tg littermates. (**D**) Neurogenesis was evaluated by immunofluorescence for 5-bromo-2'-deoxyuridine (BrdU) (red) and doublecortin (DCX, green), in which double-positive cells (yellow) were regarded as newly generated neurons. Pwd-ZSS significantly enhanced neurogenesis in the dentate gyrus (DG) and SN to a level higher than that in non-Tg littermates. (**E**) Pwd-ZSS was administered to 6- to 7-month-old Huα-Syn(A53T) mice at 0.1 mg/shot for 1 month. Mouse memory was significantly improved to a level similar to or slightly less than that of non-Tg littermates. Each point and bar represent the mean ± SEM. The numbers of total, male, and female mice analyzed are shown in each figure as n=x: m (male)+f (female).

powder on neuropathology, cognitive and motor function, oxidative stress, and cellular senescence may be attributed, to a greater or lesser extent, to the action of dietary fibers on gut microbiota. ZSS extraction residue is also expected to contain a significant amount of dietary fibers, which may account for its nootropic activity.

The present study showed that ZSS has a broad spectrum against Aβ, tau, and α-synuclein, but it remains to be studied whether ZSS is also effective against TDP-43, another protein accumulated in FTD and amyotrophic lateral sclerosis. Furthermore, we used three different mouse models of neurodegenerative diseases, but they do not fully represent human pathology, and, therefore, clinical studies are required to evaluate the true efficacy of ZSS in humans. ZSS has a long history of traditional Chinese medicine and can be treated as a non-pharmaceutical in Japan, suggesting its safety. However, since simple crushing is a different processing method than usual, the safety of the long-term intake of ZSS powder needs to be confirmed. Our results suggest that ZSS powder has antiaging effects. Twelve hallmarks of aging have been proposed, including cellular senescence, stem cell exhaustion, epigenetic alterations, loss of proteostasis, chronic inflammation, dysbiosis, and so on (*López-Otín et al., 2023*). By clarifying which of these hallmarks, in addition to cellular senescence, are improved by ZSS powder, the antiaging effects of the powder can be more clearly understood. Although we don't know the true active components in ZSS powder, and their identification may be necessary to develop ZSS-derived functional foods, our findings suggest that ZSS simple crush powder is a promising dietary material for the prevention of neurodegenerative diseases and brain aging.

## Materials and methods
### Preparation of hot water extract, simple crush powder, and extraction residue of ZSS

Dried ZSS (Origin: Hebei Province, China) was obtained from Auropure LifeScience Co, Ltd. (Zhuzhou, Hunan Province, China). The hot water extract was prepared by the company. Dried ZSS was crushed and added to 5 volumes (vol/wt) of water. These suspensions were boiled for 1 hr and passed through a filter. As an excipient, dextrin was added to the filtrates at a ratio of 80 parts solid matter to 20 parts dextrin. The mixtures were spray-dried and put through a 60-mesh sieve. Finally, the passed-through materials were collected as hot water extract. The non-extracted simple crush powder and extraction residue of ZSS were prepared in our laboratory. Dried ZSS was sterilized at 115°C for 1.5 hr, crushed using a hammer mill, and passed through a 3 mm screen. The obtained crude powder was put through a vibrating sieve (500 µm of mesh), and the passed-through material was collected as simple crush powder. The simple crush powder was added to 14 volumes of water, heated at 90°C for 3 hr, and put through a 5 µm filter. The residue on the filter was dried at 40°C under reduced pressure and collected as extraction residue.

### Component analysis of ZSS

Quantification of jujuboside A, jujuboside B, and spinosin in our ZSS preparations was outsourced to Japan Food Research Laboratories (Tokyo, Japan). Briefly, 0.2 g of materials was suspended in 30 mL of 50% methanol and shaken for 10 min. After centrifugation, the supernatant was harvested, and the pellet was subjected to methanol extraction two more times. The supernatants of three extractions were combined to a total volume of 100 mL. To detect jujuboside A and B, the extracts were 10-fold diluted and separated by high-performance liquid chromatography (HPLC) using a reverse-phase InertSustain C18 column (GL Sciences, Tokyo, Japan) with 0.1% formic acid and acetonitrile mixture (63:37) as the mobile phase. Each fraction was sequentially analyzed by electrospray ionization-mass

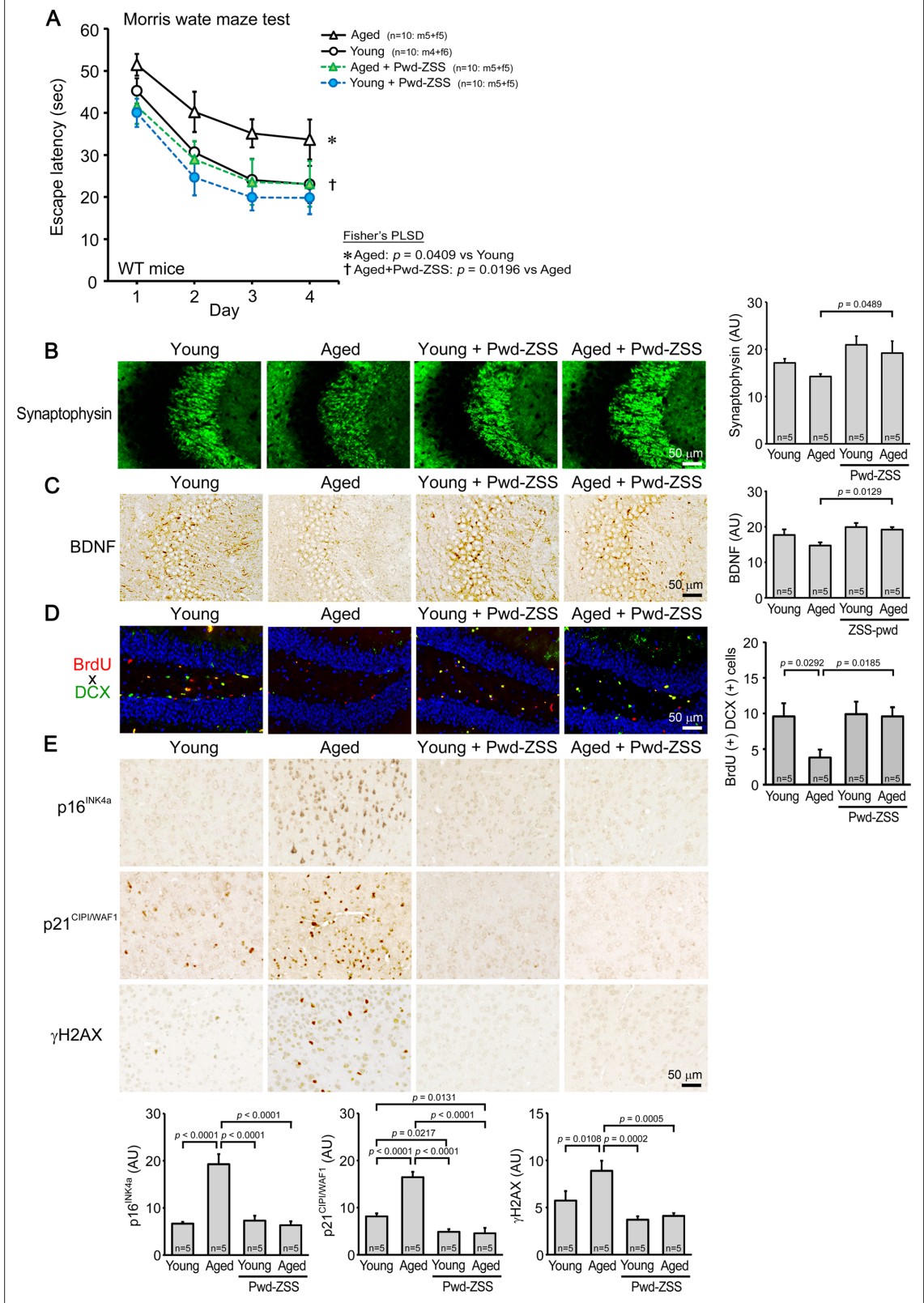

**Figure 5.** Effects of zizyphi spinosi semen (ZSS) simple crush powder on normal aged mice. Non-extracted simple crush powder of ZSS (Pwd-ZSS) was administered to 16- to 18-month-old (aged) and 8-month-old (young) wild-type mice at 0.1 mg/shot for 1 month. (**A**) Pwd-ZSS significantly improved the memory of aged mice to that of water-treated young mice. A memory-enhancing effect was also observed in young mice, but it was not significant. Pwd-ZSS significantly increased the levels of synaptophysin in the hippocampal CA2/3 regions (**B**), brain-derived neurotrophic factor (BDNF) expression

*Figure 5 continued on next page*

*Figure 5 continued*

in the hippocampus (**C**), and neurogenesis in the dentate gyrus (**D**) of aged mice to levels similar to those in water-treated young mice. In (**D**), blue shows nuclei stained with DAPI. (**E**) The levels of the cellular senescence markers p16[INK4a], p21[CIP1/WAF1], and γH2AX in the cerebral cortex of aged mice were significantly decreased by Pwd-ZSS treatment to levels similar to those in water-treated young mice. Each point and bar represent the mean ± SEM. The numbers of total, male, and female mice analyzed are shown in each figure as n=x: m (male)+f (female).

spectrometry using a Xevo TQ MS (Waters Corporation, Milford, MA, USA). For spinosin, the extracts were separated by HPLC using a reverse-phase Unison UK-C18 column (Imtakt USA, Portland, OR, USA) with 0.1% formic acid and methanol mixture (65:35) as the mobile phase. The absorbance at 270 nm of each fraction was measured.

## Mice

Three different mouse models of neurodegenerative diseases, APP23 (*Sturchler-Pierrat et al., 1997*), Tau784 (*Umeda et al., 2013*), and Huα-Syn(A53T) mice G2-3 line (*Lee et al., 2002*), were used. APP23 mice were kindly provided by Novartis Pharma, Inc, Tau784 mice were generated in our laboratory, and Huα-Syn(A53T) mice were purchased from the Jackson Laboratory (Bar Harbor, ME, USA). All Tg mice were maintained and used as heterozygotes. The mice were individually housed, and after reaching an age at which neuropathology and cognitive/motor deficits could be reliably observed, they were divided into several groups with equal mean body weight and equal number of males and females. In experiments to investigate the antiaging effects of ZSS, old and young C57BL/6 wild-type mice were used.

## Treatment of mice

Powdered materials of hot water extract, extraction residue, and non-extracted simple crush powder of ZSS were suspended in water at 0.33 or 1.65 mg/mL by sonication. 300 μL of each suspension (i.e. 0.1 or 0.5 mg material) was orally administered using feeding needles to male and female APP23, Tau784, Huα-Syn(A53T), and wild-type mice 5 days (Monday through Friday) a week for 1 month. The dose of ZSS was determined through preliminary experiments. To the control mice, 300 μL of water was orally administered for 1 month. To test the mixture of three components of ZSS, jujuboside A, jujuboside B, and spinosin (all from Biosynth, Compton, Berkshire, UK) were combined in water at concentrations of 1.52, 0.667, and 2.33 μg/mL, respectively. 300 μL of the solution was administered to Tau784 mice for 1 month. These dosages correspond to their amounts in 0.5 mg hot water extract of ZSS: i.e., 0.455 μg jujuboside A, 0.2 μg jujuboside B, and 0.7 μg spinosin. The treatments were continued during the behavioral tests. Each experiment was performed once.

## Behavioral test

The spatial reference memory of mice was examined using the Morris water maze, as described previously (*Umeda et al., 2024a*). Mice were trained to swim to a hidden platform five times a day at 5 min intervals over 4 consecutive days. The time when mice climbed on the platform was recorded. The mean time of the five trials was calculated each day. The motor function of mice was assessed by the rotarod test using an MK-610A rotarod treadmill for mice (Muromachi Kikai, Tokyo, Japan), as described previously (*Umeda et al., 2021a*). The mice were trained to stay on the rod rotating at 5 rpm for 180 s and then on the rod rotating at speed accelerated from 4 to 40 rpm over 240 s. On the next day, the accelerating rotarod test was performed two times with a 1 hr interval. The times when the mice fell off the rod were recorded. The mean time of the two trials was calculated. The behavioral tests were performed under unblinded conditions: the experimenter knew which mice were treated with ZSS.

## Histological analysis of neuropathology, cellular senescence, BDNF, and neurogenesis

After the behavioral tests, the mice in each group were divided into two groups, one for histological analysis and the other for biochemical analysis. Brain sections were prepared and stained as described previously (*Umeda et al., 2024a*). Neuropathology was examined with the following antibodies: AT8 (Thermo Scientific, Waltham, MA, USA) for phosphorylated tau, TOMA-1 (Merck-Millipore, Darmstadt, Germany) for tau oligomers, 11A1 (IBL, Fujioka, Japan) for Aβ oligomers, β001 (made in our

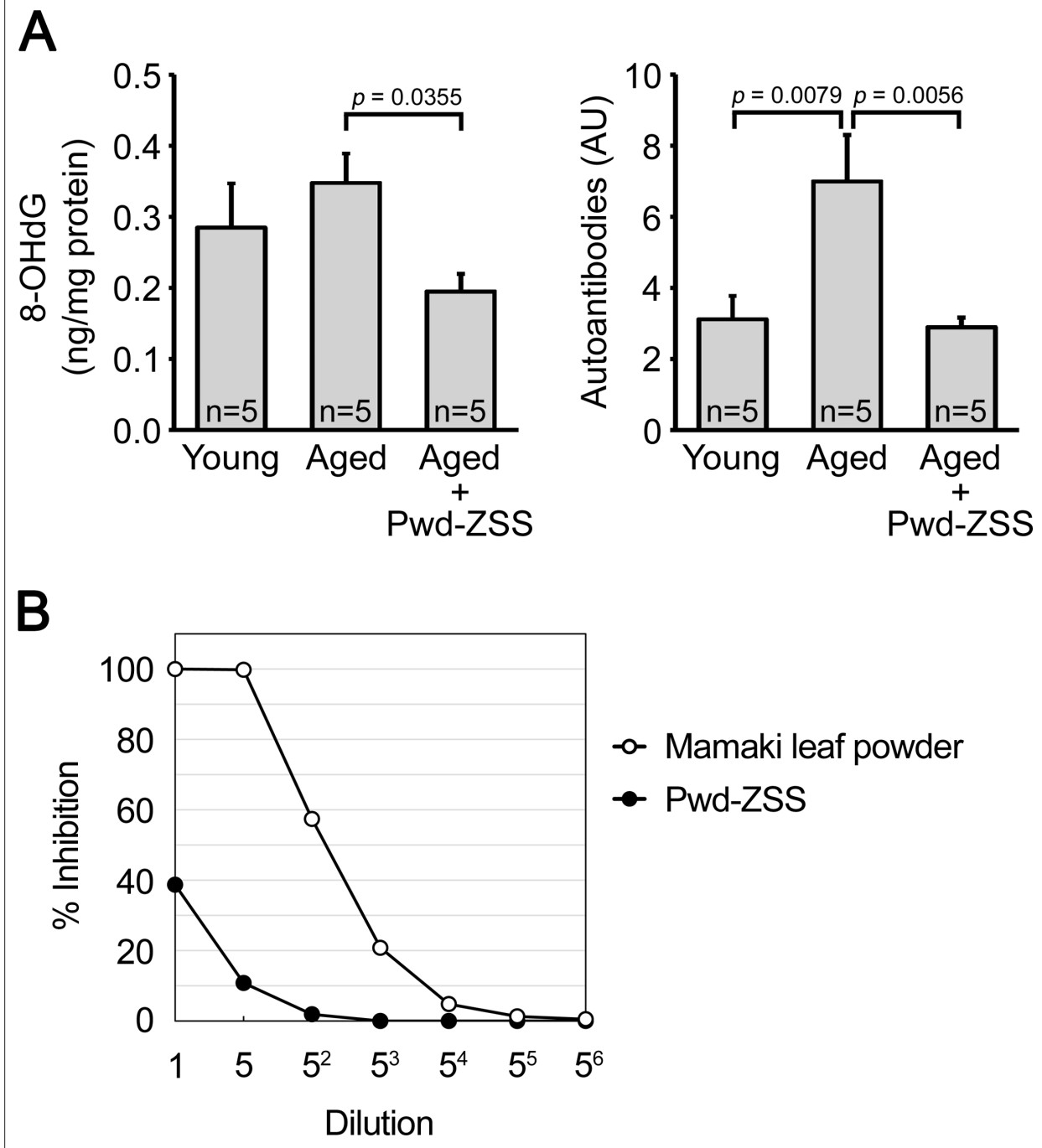

**Figure 6.** Antioxidant activities of zizyphi spinosi semen (ZSS) simple crush powder. (**A**) Antioxidant effect of ZSS powder was investigated in the brains of mice used in *Figure 5*. The levels of the DNA oxidation product 8-hydroxy-2'-deoxyguanosine (8-OHdG) and its autoantibodies were both increased in aged mice, but their levels were significantly reduced by ZSS powder to levels similar to or lower than those in young mice. Each bar represents the mean ± SEM. The numbers of mice analyzed are shown in each figure as n=x. AU, arbitrary unit. (**B**) The radical-scavenging ability of ZSS powder was measured in a cell-free system, in which the superoxide dismutase (SOD)-like activity is expressed as % inhibition against superoxide produced by xanthine oxidase. Compared to Mamaki leaf powder, which is known to have strong antioxidant activity, ZSS powder showed only a weak effect.

lab) for amyloid deposits, EP1536Y (Abcam, Cambridge, UK) for S129-phosphorylated α-synuclein, Syn33 (Sigma-Aldrich, St Louis, MO, USA) for α-synuclein oligomers, and SVP-38 (Sigma-Aldrich) for synaptophysin. Markers of cellular senescence were stained with the following antibodies: ab189034 (Abcam) for p16$^{INK4a}$, 10355-1-AP (Proteintech, Rosemont, IL, USA) for p21$^{CIP1/WAF1}$, and ab11174 (Abcam) for γH2AX. The expression of BDNF was detected using the BDNF-#9 antibody (DSHB, Iowa

**Table 1.** The three major components in 100 g of each zizyphi spinosi semen (ZSS) preparation.

| Component | Hot water extract | Simple crush powder |
| --- | --- | --- |
| Jujuboside A | 91 mg | 44 mg |
| Jujuboside B | 40 mg | 31 mg |
| Spinosin | 140 mg | 68 mg |

City, IA, USA). The staining intensity or positive area in a constant brain region was quantified using NIH ImageJ software. Neurogenesis was assessed as described previously (*Umeda et al., 2024b*); in brief, 5-bromo-2′-deoxyuridine (BrdU; Sigma-Aldrich) was intraperitoneally injected into the mice for the last 5 days of ZSS treatment. Brain sections were stained with anti-BrdU (IBL) and anti-doublecortin (DCX) antibodies (Abcam), and positive cells for both BrdU and DCX were regarded as newly generated neurons and counted in a constant brain region. Measurements from the same regions of the left and right hemisphere were averaged for each mouse.

## Biochemical analysis of DNA oxidation

Oxidative stress in mouse brain was quantified by measuring the levels of 8-OHdG, a marker of DNA oxidation caused by ROS, using the Highly Sensitive ELISA kit for 8-OHdG (Japan Institute for the Control of Aging, Fukuroi, Japan). 8-OHdG formed on chromosomal and mitochondrial DNA is excised by the action of repair enzymes and released from the cell. Brain tissues were homogenized by sonication at 100 mg wet tissue/mL in PBS containing protease inhibitor cocktail and centrifuged at $1000 \times g$ for 5 min at 4°C. The supernatants were collected and their protein content was measured. Then, the supernatants were separated into two fractions, 8-OHdG fraction and its autoantibody fraction, using the Amicon Ultra Centrifugal 100 kDa MWCO Filters (Merck Millipore, Darmstadt, Germany). The filtrates containing 8-OHdG were mixed with the primary antibody and applied to the antigen plate. The concentrated samples on the filter, which contain autoantibodies to 8-OHdG, were recovered, diluted with PBS, and allowed to react with the antigen plate in the absence of the primary

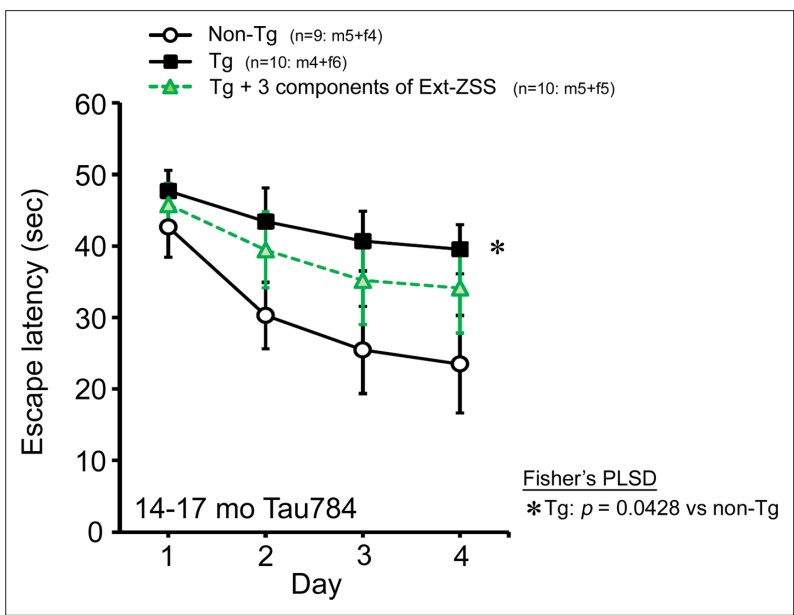

**Figure 7.** Effects of three components of zizyphi spinosi semen (ZSS) on Tau784 mice. A mixture of jujuboside A, jujuboside B, and spinosin was administered to 13- to 16-month-old Tau784 mice for 1 month. The daily doses were 0.455, 0.2, and 0.7 µg, respectively, which correspond to those in 0.5 mg ZSS hot water extract. This treatment displayed a much weaker effect on mouse memory than that of 0.5 mg ZSS hot water extract (*Figure 2A*). Each point represents the mean ± SEM. The numbers of total, male, and female mice analyzed are shown in each figure as n=x: m (male)+f (female).

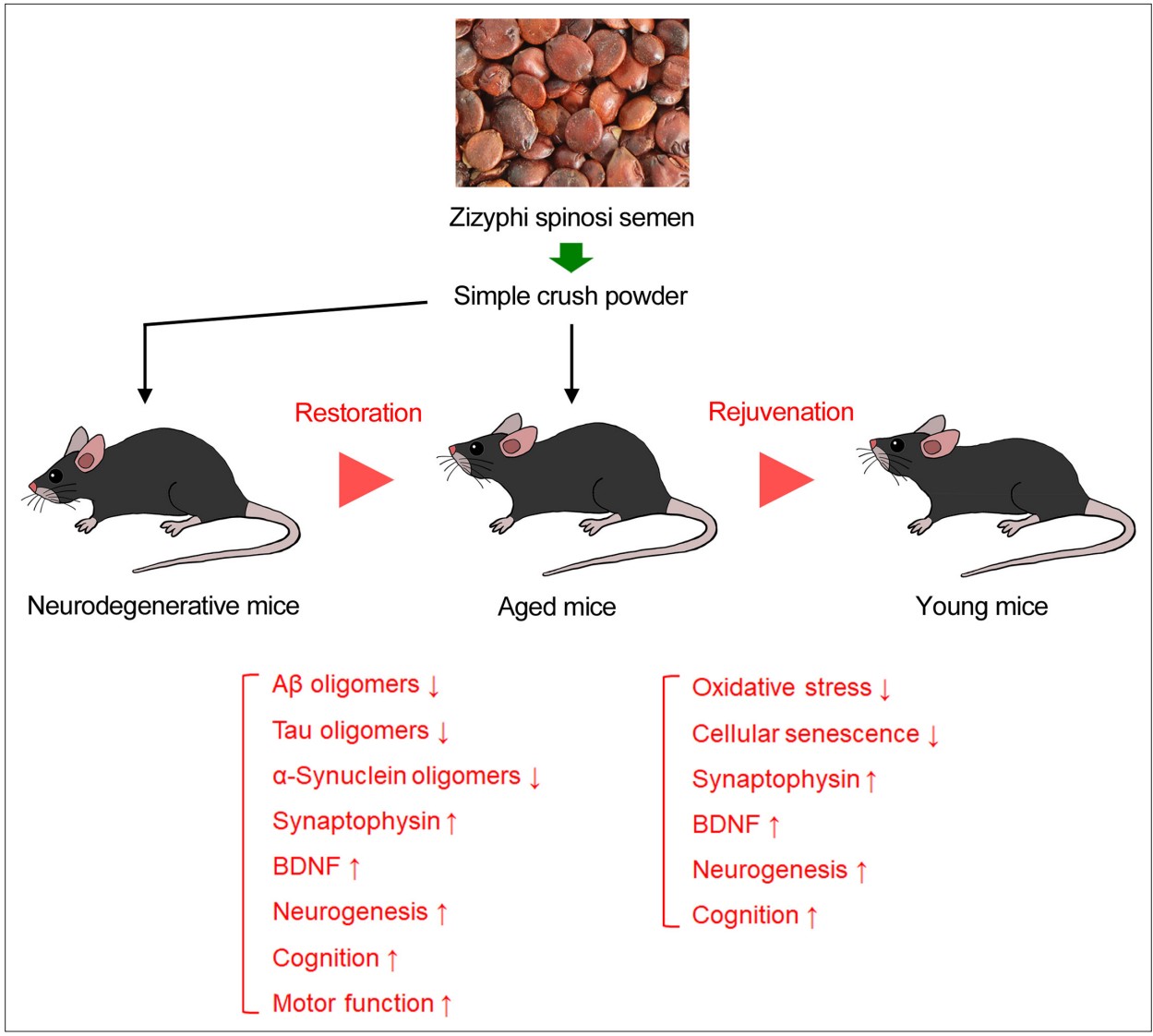

**Figure 8.** Effects of zizyphi spinosi semen (ZSS) simple crush powder on neurodegenerative diseases and brain aging. Oral administration of ZSS powder ameliorates pathological phenotypes in mouse models of neurodegenerative diseases, and, furthermore, rejuvenates the brain condition of aged mice to a level comparable to that of young mice.

antibody. Standard curve for autoantibodies was created using the primary antibody provided in the kit, with the stock concentration set at 100.

## Measurement of radical-scavenging ability of ZSS powder

Radical-scavenging ability of ZSS was measured using the SOD Assay Kit-WST (Dojindo Laboratories, Kumamoto, Japan). Superoxide dismutase (SOD)-like activity of samples against superoxide generated by xanthine oxidase was monitored with WST-1 which is reduced by superoxide and changes to WST-1 formazan to develop color. Simple crush powder of ZSS was solubilized in saline at 10 mg/mL by sonication. As a positive control, simple crush powder of Mamaki leaves, which is known to have strong antioxidant activity, was used. After centrifugation at 2000 × *g* for 5 min at 4°C, the supernatants were passed through 0.45 μm filter. The filtrates were allowed to react with WST and enzyme working solutions included in the kit. Inhibition curves were created from the absorbance at 450 nm, and SOD-like activity in the samples was calculated.

## Statistical analysis

All experiments and data analyses were performed under unblinded, open-label conditions. Comparisons of means among more than two groups were performed using ANOVA or two-factor repeated measures ANOVA (for the behavioral tests), followed by Fisher's PLSD test. Differences with a p value of <0.05 were considered significant.

## Acknowledgements

We thank Katsura Miyahara, Keigendo Pharmacy, for professional advice on traditional Chinese medicine; Ayumi Yokota, Yu Masumoto, Yuki Kinjo, Momoko Yoshida, and Miki Tsutsui for technical assistance; and Peter Karagiannis for reading the manuscript. This study was supported by funding from Cerebro Pharma Inc and from Teijin Ltd; the grant numbers are not applicable.

## Additional information

### Competing interests

Tomohiro Umeda: is an employee of Cerebro Pharma Inc, which funded this study and applied for a patent on ZSS. (PCT/JP2023/046811). Ayumi Sakai: was an employee of Cerebro Pharma Inc, which funded this study and applied for a patent on ZSS. (PCT/JP2023/046811). Ryota Nakajima: is an employee of Teijin Ltd., which funded this study and applied for a patent on ZSS. (PCT/JP2023/046811). Kei Yamana: He is an employee of Teijin Ltd., which funded this study and applied for a patent on ZSS. (PCT/JP2023/046811). Takami Tomiyama: is a founder of Cerebro Pharma Inc, which funded this study and applied for a patent on ZSS (PCT/JP2023/046811). The other authors declare that no competing interests exist.

### Funding

| Funder | Grant reference number | Author |
|---|---|---|
| Teijin Ltd. | | Takami Tomiyama |
| Cerebro Pharma Inc | | Takami Tomiyama |

The funders had no role in study design, data collection and interpretation, or the decision to submit the work for publication.

### Author contributions

Tomohiro Umeda, Formal analysis, Validation, Investigation, Visualization, Methodology; Ayumi Sakai, Rumi Uekado, Keiko Shigemori, Investigation; Ryota Nakajima, Conceptualization, Resources, Writing – review and editing; Kei Yamana, Conceptualization, Resources, Supervision, Writing – review and editing; Takami Tomiyama, Conceptualization, Supervision, Funding acquisition, Validation, Investigation, Writing – original draft, Project administration, Writing – review and editing

### Author ORCIDs

Takami Tomiyama (iD) https://orcid.org/0000-0001-6040-0178

### Ethics

All animal experiments were approved by the ethics committee of Osaka Metropolitan University and performed in accordance with the Guide for Animal Experimentation, Osaka Metropolitan University; the approval codes are 16007 (approved on 30 June 2016) and 21029 (approved 25 March 2021).

Reviewer #1 (Public review): https://doi.org/10.7554/eLife.100737.2.sa1
Reviewer #2 (Public review): https://doi.org/10.7554/eLife.100737.2.sa2
Reviewer #3 (Public review): https://doi.org/10.7554/eLife.100737.2.sa3
Author response https://doi.org/10.7554/eLife.100737.2.sa4

## Additional files

### Supplementary files
MDAR checklist

### Data availability
Data is available on Dryad.

The following dataset was generated:

| Author(s) | Year | Dataset title | Dataset URL | Database and Identifier |
|---|---|---|---|---|
| Umeda T, Sakai A, Uekado R, Shigemori K, Nakajima R, Yamana K, Tomiyama T | 2025 | Simply crushed Zizyphi spinosi semen prevents neurodegenerative diseases and reverses age-related cognitive decline in mice | http://doi.org/10.5061/dryad.vt4b8gv43 | Dryad Digital Repository, 10.5061/dryad.vt4b8gv43 |

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
