## [Editor Report · eLife assessment]

The authors made a **useful** finding that Zizyphi spinosi semen, a traditional Chinese medicine, has demonstrated excellent biological activity and potential therapeutic effects against Alzheimer's disease (AD). The researchers presented the effects, but the research evidence for the mechanism was **incomplete**. The main claims were only partially supported.

---

## [Referee Report · Reviewer #1 (Public review)]

Summary:

The study shows that Zizyphi spinosi semen (ZSS), particularly its non-extracted simple crush powder, has significant therapeutic effects on neurodegenerative diseases. It removes Aβ, tau, and α-synuclein oligomers, restores synaptophysin levels, enhances BDNF expression and neurogenesis, and improves cognitive and motor functions in mouse AD, FTD, DLB, and PD models. Additionally, ZSS powder reduces DNA oxidation and cellular senescence in normal-aged mice, increases synaptophysin, BDNF, and neurogenesis, and enhances cognition to levels comparable to young mice.

Weaknesses:

(1) While the study demonstrates that ZSS has protective effects across a wide range of animal models, including AD, FTD, DLB, PD, and both young and aged mice, it is broad and lacks a detailed investigation into the underlying mechanisms. This is the most significant concern.

(2) The authors highlight that the non-extracted simple crush powder of ZSS shows more substantial effects than its hot water extract and extraction residue. However, the manuscript provides very limited data comparing the effects of these three extracts.

(3) The authors have not provided a rationale for the dosing concentrations used, nor have they tested the effects of the treatment in normal mice to verify its impact under physiological conditions.

(4) Regarding the assessment of cognitive function in mice, the authors only utilized the Morris Water Maze (MWM) test, which includes a five-day spatial learning training phase followed by a probe trial. The authors focused solely on the learning phase. However, it is relevant to note that data from the learning phase primarily reflects the learning ability of the mice, while the probe trial is more indicative of memory. Therefore, it is essential that probe trial data be included for a more comprehensive analysis. A justification should be included to explain why the latency of 1st is about 50s not 60s.

(5) The BDNF immunohistochemical staining in the manuscript appears to be non-specific.

(6) The central pathological regions in PD are the substantia nigra and striatum. Please replace the staining results from the cortex and hippocampus with those from these regions in the PD model.

---

## [Referee Report · Reviewer #2 (Public review)]

Summary:

The authors studied the effects of hot water extract, extraction residue, and non-extracted simple crush powder of ZSS in diseased or aged mice. It was found that ZSS played an anti-neurodegenerative role by removing toxic proteins, repairing damaged neurons, and inhibiting cell senescence.

Strengths:

The authors studied the effects of ZSS in different transgenic mice and analyzed the different states of ZSS and the effects of different components.

Weaknesses:

The authors' study lacked an in-depth exploration of mechanisms, including changes in intracellular signal transduction, drug targets, and drug toxicity detection.

---

## [Referee Report · Reviewer #3 (Public review)]

ZSS has been widely used in Traditional Chinese Medicine as a sleep-promoting herb. This study tests the effects of ZSS powder and extracts on AD, PD, and aging, and broad protective effects were revealed in mice.

However, this work did not include a mechanistic study or target data on ZSS were included, and PK data were also not involved. Mechanisms or targets and PK study are suggested. A human PK study is preferred over mice or rats. E.g. which main active ingredients and the concentration in plasma, in this context, to study the pharmacological mechanisms of ZSS.

---

## [Author Response]

**Public Reviews:**

**Reviewer #1 (Public review):**
(1) While the study demonstrates that ZSS has protective effects across a wide range of animal models, including AD, FTD, DLB, PD, and both young and aged mice, it is broad and lacks a detailed investigation into the underlying mechanisms. This is the most significant concern.

We appreciate this comment. We recognize that elucidating the mechanism is an important research topic, and we are currently working on it. The purpose of publishing this paper at this time is to inform the public as soon as possible about natural materials and methods that may be effective in preventing dementia and neurodegenerative diseases, and to encourage similar research.

(2) The authors highlight that the non-extracted simple crush powder of ZSS shows more substantial effects than its hot water extract and extraction residue. However, the manuscript provides very limited data comparing the effects of these three extracts.

Certainly, it would be better to compare them in several different models, but we believe that important results have already been obtained in tau Tg mice, and comparative data in other models are just additive and confirmatory.

(3) The authors have not provided a rationale for the dosing concentrations used, nor have they tested the effects of the treatment in normal mice to verify its impact under physiological conditions.

As described in the Materials and Methods section, the dosage was determined based on the results of preliminary experiments. The beneficial effects in normal mice are shown in Figure 5.

(4) Regarding the assessment of cognitive function in mice, the authors only utilized the Morris Water Maze (MWM) test, which includes a five-day spatial learning training phase followed by a probe trial. The authors focused solely on the learning phase. However, it is relevant to note that data from the learning phase primarily reflects the learning ability of the mice, while the probe trial is more indicative of memory. Therefore, it is essential that probe trial data be included for a more comprehensive analysis. A justification should be included to explain why the latency of 1st is about 50s not 60s.

We agree that it is better to include the results of the probe test. We did not include them this time, but we would like to include them in the future. In the memory acquisition training, five trials were performed per day. Since the mice learned the location of the platform during the first five trials, the latency on the first day became around 50 seconds.

(5) The BDNF immunohistochemical staining in the manuscript appears to be non-specific.

We cannot understand the basis for saying it is non-specific.

(6) The central pathological regions in PD are the substantia nigra and striatum. Please replace the staining results from the cortex and hippocampus with those from these regions in the PD model.

We examined the substantia nigra and found that synuclein pathology appeared in Tg mice and was suppressed by ZSS administration. However, because we did not investigate the striatum, we decided not to show the results for the nigrostriatal system this time. Instead, we thought that we could demonstrate the inhibitory effect of ZSS on synuclein pathology by showing the results for the hippocampus, which showed early functional decline in these mice (Fig. 4E).

**Reviewer #2 (Public review):**
The authors' study lacked an in-depth exploration of mechanisms, including changes in intracellular signal transduction, drug targets, and drug toxicity detection.

We appreciate this comment. We understand that the mechanism, targets, and toxicity are important issues to be considered in the future.

**Reviewer #3 (Public review):**
However, this work did not include a mechanistic study or target data on ZSS were included, and PK data were also not involved. Mechanisms or targets and PK study are suggested. A human PK study is preferred over mice or rats. E.g. which main active ingredients and the concentration in plasma, in this context, to study the pharmacological mechanisms of ZSS.

We appreciate this comment. We understand that the mechanism and target are important issues to consider in the future. As the reviewer pointed out, to conduct PK studies, we must first identify the active ingredients. Unfortunately, we have not been able to identify them yet.

**Reviewer #2 (Recommendations for the authors):**
The authors have proved that ZSS has neuroprotective effects through rigorous animal experiments. However, ZSS contains other active substances besides jujuboside A, jujuboside B, and spinosin, which is more concerning. More critical data may be obtained if experiments have been designed to search for active substances.

We appreciate this suggestion. We recognize that identifying the true active ingredients is a very important issue. Future studies will be designed to identify them and elucidate their mechanism of action.